# Is the Course of COVID-19 Different during Pregnancy? A Retrospective Comparative Study

**DOI:** 10.3390/ijerph182212011

**Published:** 2021-11-16

**Authors:** Marcin Januszewski, Laura Ziuzia-Januszewska, Alicja A. Jakimiuk, Waldemar Wierzba, Anna Głuszko, Joanna Żytyńska-Daniluk, Artur J. Jakimiuk

**Affiliations:** 1Department of Obstetrics and Gynecology, Central Clinical Hospital of the Ministry of Interior and Administration, 02-507 Warsaw, Poland; lek.med.mjanuszewski@gmail.com (M.J.); waldemar.wierzba@cskmswia.gov.pl (W.W.); 2Department of Otolaryngology, Central Clinical Hospital of the Ministry of Interior and Administration, 02-507 Warsaw, Poland; lauraziuzia@gmail.com; 3Department of Plastic Surgery, Central Clinical Hospital of the Ministry of Interior and Administration, 02-507 Warsaw, Poland; alajakimiuk@hotmail.com; 4Satellite Campus in Warsaw, University of Humanities and Economics, 01-513 Warsaw, Poland; 5Department of Neonatology, Central Clinical Hospital of the Ministry of Interior and Administration, 02-507 Warsaw, Poland; anna.gluszko@cskmswia.pl (A.G.); joanna.daniluk@cskmswia.pl (J.Ż.-D.); 6Center for Reproductive Health, Institute of Mother and Child, 01-211 Warsaw, Poland

**Keywords:** SARS-CoV-2, COVID-19, clinical course, pregnancy, oxygen flow, inflammatory markers, ferritin, laboratory indicators

## Abstract

The COVID-19 pandemic has challenged health systems around the world. Maternal-foetal medicine, which has been particularly affected, must consider scientific data on the physiological processes occurring in the pregnant woman’s body to develop relevant standards of care. Our study retrospectively compared the clinical and laboratory characteristics of 52 COVID-19 pregnant patients with 53 controls. Most of the pregnant patients required medical attention during the third trimester and therefore we propose that vaccination is needed prior to the 30th week of pregnancy. We found no differences between the 2 groups in the course of illness classification system, days of hospital stay, need for oxygen supplementation, need for mechanical ventilation, and ICU admission. Moreover, clinical manifestations and imaging findings were comparable. Pregnant patients needed a greater oxygen flow rate and required high flow oxygen therapy more frequently. Considering pregnancy-related physiological adaptations, we found that COVID-19 infection in pregnant patients is associated with higher levels of inflammatory markers, apart from serum ferritin, than in non-pregnant women, and concluded that biomarkers of cardiac and muscle injury, as well as kidney function, may not be good predictors of COVID-19 clinical course in pregnant patients at the time of admission, but more research needs to be conducted on this topic.

## 1. Introduction

Coronavirus disease 2019 (COVID-19), caused by severe acute respiratory syndrome coronavirus 2 (SARS-CoV-2), was declared a pandemic by WHO on 11 March 2020, and has exposed vulnerable populations to an unprecedented global health crisis. Maternal-foetal medicine was unprepared and struggled to mount an adequate response. As new data on SARS-CoV-2 are emerging, guidelines regarding perinatal care are constantly being updated to balance evidence-based maternity care with COVID-19 management and treatment strategies.

The SARS-CoV-2 infection mainly affects the respiratory system, causing mild or moderate respiratory symptoms in 85% of cases. Cardiovascular, renal, neurological, psychiatric, dermatological, and gastrointestinal manifestations have also been reported [1,2,3,4,5]. Guan et al., [2] reported fever (88.7%), cough (67.8%), fatigue (38%), sputum production (33.7%), shortness of breath (18.7%), myalgia or arthralgia (14.9%), sore throat (13.9%), headache (13.6%), chills (11.5%), nausea or vomiting (5%), nasal congestion (4.8%), and diarrhea (3.85%) as the leading symptoms. In 5–30% of patients the virus causes severe acute respiratory syndrome (SARS) which leads to the use of mechanical ventilation methods and can progress to multiorgan failure [1,2,3]. Acute respiratory distress syndrome (ARDS) is represented by hypoxemic respiratory failure with bilateral lung infiltrates. Almost 5% of patients with COVID-19 develop a severe form of the disorder that requires treatment in an intensive care unit (ICU) [6,7,8]. The frequency of progression towards greater clinical severity and the likelihood of ICU admission in COVID-19 patients is greater with increasing age, male sex, obesity, and comorbidities such as hypertension, asthma, diabetes, and cardiovascular diseases [1,2,3]. According to Lauer et al., [9] the median incubation period, the time between exposure to the virus and the appearance of symptoms, was estimated to be 5.1 days, and 97.5% of those who develop symptoms will do so within 11.5 days.

The fact that SARS-CoV-2 has the potential to spread to extrapulmonary tissues has an impact on its clinical manifestations and is based on the binding and entry mechanism into the host cell [10].

The spike (S) protein of coronaviruses enables viral entry into target cells. Entry depends on the binding of the surface unit of the S protein, S1, to a cellular receptor, the human receptor angiotensin I-converting enzyme 2 (ACE2) [11], which facilitates viral attachment to the surface of target cells. Entry requires S protein activation by cellular proteases. In this process, driven by the S2 subunit and mediated by cellular serine protease (TMPRSS2), the fusion of viral and cellular membranes occurs [12,13,14].

SARS-CoV-2 virulency includes direct viral cytopathic effect and induction of a spontaneous immune response characterized by an exaggerated production of soluble immune mediators. This phenomenon has been called a “cytokine storm”, and as a dysregulated inflammatory reaction, it is responsible for mediating tissue damage in patients with COVID-19 [15,16,17,18]. These states of constant inflammation and hypercoagulation interplay, creating an endless feedback loop [19].

Pregnancy increases the severity of respiratory tract infections such as influenza [20,21], or RSV [22], due to physiological adaptations to pregnancy, such as the higher diaphragm position that prompts a restriction in lung expansion, increased oxygen demand, and progesterone-related airway swelling [23]. Intolerance to hypoxemia predisposes women to complications associated with respiratory infections, leading to maternal and foetal mortality and morbidity [24,25].

In addition, an immunologic modulation occurs, from a pro-inflammatory state (profitable for cell clearance, angiogenesis, and foetal growth) during the first trimester, to an anti-inflammatory condition combined with skewing towards humoral immunity (favourable for foetal growth) in the second trimester, and finally reaching a second pro-inflammatory state during the third trimester (initiation of parturition) [26,27].

Pregnant women’s sensitivity to infections and the hypercoagulation state places them in a risk group for COVID-19 infection and the increased risk of pregnancy-related complications such as miscarriages, premature birth, preeclampsia, and foetal growth restriction. Pregnancy loss, especially when occurring during the first trimester in SARS-CoV-2 infected women, is probably due to placenta infection and insufficiency [28]. Higher rates of preeclampsia were observed in a study by Conde-Agudelo et al. [29]. Di Mascio D et al., and Wei SQ et al., found increased incidences of both preeclampsia and premature births in SARS-CoV-2 infected pregnancies [30,31]. Moreover, the latter study also suggested there was an association between low birth weight and the severe form of COVID-19 [31].

Moreover, serious adverse outcomes among pregnant women with previous SARS and MERS coronavirus infections have been observed, including maternal and neonatal deaths, preterm birth, and neonatal intensive care treatment [32].

The objective of our study was to compare the clinical course and laboratory findings of pregnant women during their second and third trimesters, who were infected with SARS-CoV-2, with a cohort of non-pregnant women of reproductive age with a confirmed diagnosis of COVID-19. The outcomes we measured in relation to the severity of the disease were: course of illness classification system, increased demand for oxygen supplementation, days of hospital stay, need for oxygen supplementation and high flow oxygen therapy, need for mechanical ventilation, and ICU admissions.

## 2. Materials and Methods

### 2.1. Study Population

This retrospective single-centered study was carried out in the Department of Obstetrics and Gynecology, at the Central Clinical Hospital of the Ministry of the Interior and Administration in Warsaw, Poland. The control group was comprised of non-pregnant women of reproductive age who were randomly recruited from those designated for COVID-19 treatment in Hospital Departments. From 15 May 2020 to 26 April 2021, a total of 52 pregnant and 53 non-pregnant women with COVID-19 infection were admitted for treatment. The research was approved by the Bioethics Committee of the Central Clinical Hospital of Interior and Administration in Warsaw.

### 2.2. Inclusion and Exclusion Criteria

Inclusion criteria were temperature >39 °C despite the use of paracetamol, tachypnoea >30/min, SaO_2_ <95% on room air, oxygen requirement, or critical course of illness. The COVID-19 diagnosis was confirmed by positive RT-PCR assay performed within no more than 13 days prior to admission.

Exclusion criteria were hospital admissions for problems other than those reflecting a deteriorating condition.

### 2.3. Clinical Course of Illness

We divided the patients into 4 groups based on the Polish Association of Epidemiologists and Infectiologists [33] classification guidelines determined by the severity of symptoms and the results of tests. Mild (asymptomatic, or the presence of cough, fever, dyspnoea, fatigue, headache, muscle pains, nausea, vomiting, diarrhoea), moderate (clinical and radiological features of lung occupation), severe (respiratory failure, low peripheral SpO_2_ < 90%), or critical (ARDS, hypotensive shock, multiorgan failure, loss of consciousness).

### 2.4. Study Procedures

At admission, all the women underwent blood, urine, coagulation, and biochemical blood tests, as well as computed tomography (without contrast) in cases where moderate, severe, or critical forms of COVID-19 were suspected.

The following variables were analysed: patient age, weight, height, body mass index (BMI), pre-existing medical conditions (diabetes mellitus, hypertension, hypothyroidism, asthma), symptoms, physical examination results, pregnancy status, and gestational age at the initial presentation.

### 2.5. Statistical Analysis

Statistical analysis was performed using Statistica 13.1 statistical software (StatSoft Polska Sp. z o.o., Kraków, Poland). Mean values and standard deviations were used to describe the study groups; in case of skewed distributions the median was calculated as a measure of central tendency, and the scatter was presented using the 25th and 75th percentile of the distribution; in case of qualitative variables the data were presented as a percentage. After assessing the normality of the distributions of the explanatory variables using the Shapiro–Wilk test, a comparative analysis of each group was performed. For variables that did not meet the assumption of near normal distribution, the Mann–Whitney U test was used to compare groups; for variables that met the assumption of normal distribution, the Student’s *t*-test was used for comparison. In the case of qualitative variables, the chi-squared test was used to compare the frequencies of the studied characteristics in groups. Fisher’s exact test was applied when the expected number of frequencies in the chi-square test was less than 5. Differences were considered statistically significant at *p* < 0.05.

## 3. Results

The median gestational age was 30 (IQR-7) weeks ranging from 17 to 37 weeks (Table 1). The mean age of patients was 31.9 ± 4.79 years, the median weight was 77 (IQR-26) kg, the median body mass index was 28.36 (IQR-9.88) kg/m2. Patients’ characteristics are summarized in Table 2. The main clinical symptoms were dyspnoea (n = 48, 92.31%), cough (n = 47, 90.38%) fever (n = 33, 63.46%), fatigue (n = 22, 42.31%), and muscle aches (n = 22, 42.31%). Dyspnoea and cough were more frequent, and diarrhoea less frequent, in the pregnant group (Table 3). Comorbidities included diabetes (n = 9, 17.65%), hypertension (n = 5, 10.00%), asthma (n = 2, 3.85%), and hypothyroidism (n = 18, 35.29%), the latter being more common in the pregnant cohort. (Table 4). Nine (17.31%) cases were classified as a mild course of COVID-19. Moderate, severe, and critical COVID-19 accounted for 25 (48.08%), 17 (32.69%), and 1 (1.92%) cases respectively (Table 5). The median duration of hospitalization was 8 (IQR = 6, range 2–23) days. The median length of time from the onset of SARS-CoV-2 symptoms was 7 (IQR = 6, range 1–23) days. The median percentage of lung involvement on CT was 20% (IQR = 11). Forty-two (80.77%) pregnant COVID-19 patients required oxygen supplementation and 9 (17.31%) needed high-flow therapy. The median oxygen flow rate was 4 (IQR = 4.5) l/min which was significantly higher than in the non-pregnant group. ICU admission was necessary for 2 (3.85%) cases and no mechanical ventilation was needed in the pregnant group. Clinical course parameters are presented in Table 6. Six (11.5%) cases delivered, in weeks 29, 32, 35, 36, 36, and 37 of pregnancy. Comparison of the cases’ laboratory results with those of the non-pregnant group revealed higher level of leucocytes, neutrophiles, CRP, procalcitonin, IL-6, D-dimer, and fibrinogen, and significantly lower levels of hemoglobin, lymphocytes, platelets, NT pro-BNP, calcium, ferritin, creatinine, urea, magnesium, sodium, and prothrombin time in the pregnant cohort (Table 7).

## 4. Discussion

### 4.1. Patients’ Characteristics and Comorbidities

Mean gestational age at admittance was 30 weeks. Based on current knowledge, a severe and critical COVID-19 course of illness is mainly a hyperinflammatory, immune-mediated disorder, triggered by a viral infection. As a result of their immunological features, pregnant women are supposed to be particularly susceptible to intracellular infections as well as immunological disturbances, especially during the late second and third trimesters [27]. During the second trimester, these changes are characterized mostly by the elevation of humoral immune responses and suppression of cell-mediated immunity, referred to as the T-helper lymphocyte type 1-type 2 (Th1-Th2) shift [34,35]. Therefore, Th1 cell-mediated immunity is compromised, increasing the susceptibility of pregnant women to viral and intracellular bacterial infections [26,27,36]. During the third trimester, increased numbers of monocytes and granulocytes are found in maternal blood, compared to non-pregnant women, releasing inflammatory cytokines, e.g., IL-8, TNF-a, and IL-6 [27,37,38]. Moreover, pathophysiological processes responsible for the development of hypertension, diabetes, or cardiovascular diseases intensify during the 3rd trimester and pregnant women with comorbidities and older pregnant women appear to be at particularly elevated risk of adverse maternal outcomes [23,39,40,41,42]. We believe that this should addressed by implementing preventive measures in the form of vaccination prior to week 30 of pregnancy. The use of mRNA vaccines for both maternal and foetal benefits is well established in the literature. The efficiency and safety of inoculation against SARS-CoV-2 in pregnant and lactating cohorts were underlined in studies by Collier et al., and Gray et al. [43,44]. Antibodies detected in umbilical blood and milk suggests the possibility of infant protection and their cord blood titers show a correlation with the time from vaccination to delivery [45]. Data show the presence of foetal antibodies after 15 days of both mRNA vaccine and infection. [46] However, impaired placental transmission during the third trimester has been recorded [46,47]. Therefore, late second trimester may be proposed as a perfect time for vaccination [48], although further evaluation of this is needed.

Hypothyroidism was the only comorbidity that was statistically more frequent in the pregnant group. Thyroid hormone dysfunction could affect maternal COVID-19 outcomes and increase mortality in patients with a critical illness [49,50,51]. Moreover, in a retrospective study of hospitalized moderate-to-critical COVID-19 patients, 56% of the 50 subjects studied showed lower-than-normal TSH values during their infection [52]; it has been suggested that it is necessary to monitor thyroid hormones in COVID-19 patients [53]. In our opinion however, serum TSH FT3 and FT4 concentrations in pregnant patients with a combination of primary hypothyroidism and levothyroxine treatment might be misinterpreted. Our two groups did not differ in terms of the time of SARS-CoV-2 onset, length of hospital stay, and lung involvement in the CT scan. In contrast, more severe disease at imaging has been documented in a cohort of pregnant patients from China [54].

### 4.2. Symptoms

The high prevalence of dyspnoea (n = 48, 92.31%), cough (n = 47, 90.38%), fever (n = 33, 63.46%), fatigue (n = 22, 42.31%), and muscle aches (n = 22, 42.31%), and the greater frequency of dyspnoea in pregnant group may be partially associated with the physiology of pregnancy and is consistent with the findings of Zambrano et al., [39]. Other studies among pregnant patients have revealed 73% [41] and 95% [55] were asymptomatic. Pregnant women have also been found to be more likely to be asymptomatic than non-pregnant women of reproductive age with COVID-19 [41]. However, these analyses included pregnant and recently pregnant women diagnosed as having either suspected or confirmed COVID-19 who were attending or admitted to hospital for any reason. At any given time, pregnant women account for 3% (234 million) of the world population of 7.8 billion [56] and only fraction of this cohort was tested. Therefore, it is not possible to identify with complete accuracy how many pregnant women have been affected and all pregnant persons should be monitored for development of symptoms and signs of COVID-19.

### 4.3. Clinical Course Comparison

Our case-control study, which included patients admitted to hospital for non-obstetric reasons with RT-PCR confirmed infection, indicated that pregnant women diagnosed with COVID-19 did not present more severe outcomes than their nonpregnant counterparts. Data emerging from the meta-analyses by Allotey et al., [41] Khali et al., [57] and Mazur-Bialy et al., [58] show that pregnant women may have an increased risk of developing severe symptoms. In meta-analysis made by Zambrano et al., [39] that included over 23,000 pregnant women and more than 386,000 nonpregnant women of reproductive age with symptomatic laboratory-confirmed SARS-CoV-2 infection, pregnant patients had a higher risk of ICU admission, receiving invasive ventilation, ECMO, and death. However, lack of information on the women’s pregnancy status and reasons for hospitalisation limits the usefulness of the study’s findings. Another meta-analysis made by Martinez-Portilla et al., [59] that compared 5183 pregnant with 175,905 nonpregnant women after propensity score-matching revealed a higher risk of death, pneumonia, and ICU admission.

Our study showed that the oxygen flow rate and the need for high-flow oxygen therapy were both higher in the pregnant group in comparison with the non-pregnant subjects. Due to a comparable lung involvement percentage in the 2 groups, the physiological adaptations during pregnancy (higher diaphragm position due to increased uterus size, increased oxygen demand, and progesterone-induced airway oedema) may play a role.

### 4.4. Laboratory Findings in Comparison with Non-Pregnant Group

The results of the laboratory tests performed on the pregnant group on admission comparing the inflammatory markers with those non-pregnant group revealed that the leucocyte counts, neutrophiles counts, CRP levels, procalcitonin levels, and IL-6 were higher, the exception being ferritin levels. Some findings are consistent with similar studies and may be explained by the hyper induction of the immune system caused by pregnancy and the SARS-CoV-2 infection [60]. Data from Liu et al., and Qiancheg et al., shows that an elevated leukocytes count in the blood is significantly more frequently observed among pregnant COVID-19-affected women [61,62]. Nevertheless, an analysis of the data reported by Liu et al., shows that the number of leukocytes is almost in the high normal range of values for pregnant women. Additionally, mild leukocytosis physiologically occurs in the third trimester [61,62,63]. Serum levels of ferritin were not significantly higher, probably due to the higher frequency of iron deficiency anaemia in the pregnant group [64].

In our study, the hypercoagulation state manifested in higher D-dimer and fibrinogen levels and lower prothrombin times may have been partially pregnancy-induced. Available data suggest an increased risk of venous thromboembolism (VTE) related to COVID-19 in infected pregnant patients compared with uninfected pregnant patients [65,66,67].

Our study revealed lower haemoglobin, platelets, creatinine, and urea levels typical for pregnancy resulting from hemodilution, increased renal flow rate and pregnancy-related thrombocytopaenia.

We also detected some shifts in ion levels due to the physiological changes during pregnancy, namely lower magnesium, calcium, and sodium levels.

Interestingly, we also discovered significantly lower NT-proBNP levels in the pregnant group and similarly for troponin levels. Nt-ProBNP is produced by the heart and the pregnant uterus, because of relevant cardiac volume overload in chronic heart failure. Between 8 and 28% of COVID-19 patients show evidence of cardiac injury with elevated troponin and natriuretic peptides [68,69]. Some data support NT-proBNP as an independent risk factor for in-hospital death, ICU admission, mechanical ventilation, and coagulopathy in patients with severe COVID-19 disease [70].

## 5. Conclusions

Most pregnant patients infected with SARS-CoV-2 require medical attention during the third trimester. We propose that all pregnant women should be vaccinated prior to the 30th week of pregnancy.

Pregnant women diagnosed with COVID-19 did not present more severe outcomes than their non-pregnant counterparts, probably due to the small patient sample and imperfect measuring methods. Clinical manifestations of COVID-19 disease were similar in the two cohorts, and some, like dyspnoea or fatigue, overlap with the symptoms of normal pregnancy.

The laboratory and imaging findings were generally similar in both the pregnant and non-pregnant groups. Laboratory results for the coagulation test, morphology, or serum levels of ions can be explained by normal pregnancy-related physiological processes. Moderate and severe COVID-19 clinical courses in pregnant women are associated with higher levels of inflammatory markers, apart from serum ferritin, than in non-pregnant women, which might be partially explained by the imposition of immunological features and iron deficiency anaemia. Biomarkers of cardiac and muscle injury, as well as kidney function, may not be good predictors of COVID-19 clinical course in pregnant patients at the time of admission, but more research needs to be conducted on this topic.

### 5.1. Research Implications

Future studies of pregnant patients infected with SARS-CoV-2 should focus on the assessment of disease severity predictors for early identification of patients at risk of developing complications, and thus improve optimalization and preventative efforts in this cohort. Further validation with a larger study group and risk models are needed to provide useful data for the effective management of health resources in the COVID-19 era.

### 5.2. Strengths and Limitations

The main strength of our study is the variety of parameters evaluated.

The main weaknesses of our investigation include its single-centre nature, as well as a small and homogeneous group of patients.

## Figures and Tables

**Table 1 ijerph-18-12011-t001:** Weeks of pregnancy.

hbd	n (%)
≤26	12 (23.08)
17	1 (1.92)
22	3 (5.77)
24	2 (3.85)
25	2 (3.85)
26	4 (7.69)
27–30	15 (28.85)
27	4 (7.69)
28	2 (3.85)
29	6 (11.54)
30	3 (5.77)
31–33	8 (15.38)
31	1 (1.92)
32	4 (7.69)
33	3 (5.77)
34–36	15 (28.85)
34	7 (13.46)
35	6 (11.54)
36	2 (3.85)
≥37	2 (3.85)
37	2 (3.85)

**Table 2 ijerph-18-12011-t002:** Patients’ characteristics.

Variable		Cases	Controls	*p*-Value
Age (years)	N	52	53	0.282
	Mean ± SD	31.9 ± 4.79	32.96 ± 5.22	
	Range	22–40	23–45	
hbd	N	52	0	
	Median (IQR)	30 (7)		
	Range	17–37		
Weight (kg)	N	43	33	0.566
	Median (IQR)	77 (26)	75 (27)	
	Range	54–130	52–120	
Height (cm)	N	43	33	0.415
	Mean ± SD	166.12 ± 5.83	167.21 ± 5.69	
	Range	151–175	156–180	
BMI (kg/m^2^)	N	43	35	0.379
	Median (IQR)	28.36 (9.88)	29.3 (7.71)	
	Range	20.82–45.91	18.59–45.17	

**Table 3 ijerph-18-12011-t003:** Patients’ symptoms.

Variable		Cases	Controls	*p*-Value (Pearson’s Chi-Square Test)
Fatigue	no	30 (57.69%)	29 (54.72%)	0.759 *
	yes	22 (42.31%)	24 (45.28%)	
Sputum	no	49 (94.23%)	47 (88.68%)	0.488 **
	yes	3 (5.77%)	6 (11.32%)	
**Dyspnoea**	no	4 (7.69%)	16 (30.77%)	**0.003 ***
	yes	48 (92.31%)	36 (69.23%)	
Muscle aches	no	30 (57.69%)	31 (58.49%)	0.934 *****
	yes	22 (42.31%)	22 (41.51%)	
Sore throat	no	46 (88.46%)	47 (88.68%)	0.972 *****
	yes	6 (11.54%)	6 (11.32%)	
Headache	no	40 (76.92%)	35 (66.04%)	0.217 *****
	yes	12 (23.08%)	18 (33.96%)	
Chills	no	50 (96.15%)	52 (98.11%)	0.618 **
	yes	2 (3.85%)	1 (1.89%)	
Nausea	no	50 (96.15%)	48 (90.57%)	0.437 **
	yes	2 (3.85%)	5 (9.43%)	
Nasal discharge	no	47 (90.38%)	48 (90.57%)	1 **
	yes	5 (9.62%)	5 (9.43%)	
**Diarrhoea**	no	50 (96.15%)	44 (83.02%)	**0.028 ***
	yes	2 (3.85%)	9 (16.98%)	
Smell and taste disorder	no	38 (73.08%)	34 (64.15%)	0.325 *
	yes	14 (26.92%)	19 (35.85%)	
**Cough**	no	5 (9.62%)	20 (37.74%)	**<0.001 ***
	yes	47 (90.38%)	33 (62.26%)	
Fever	no	19 (36.54%)	19 (35.85%)	0.941 *
	yes	33 (63.46%)	34 (64.15%)	

* Pearson’s chi-square test, ** Fisher’s exact test.

**Table 4 ijerph-18-12011-t004:** Patient’s comorbidities.

Variable		Cases	Controls	*p*-Value (Pearson’s Chi-Square Test)
Preeclampsia	no	50 (98.04%)	53 (100%)	0.490 **
	yes	1 (1.96%)	0 (0%)	
**Hypothyroidism**	no	33 (64.71%)	44 (83.02%)	**0.033 ***
	yes	18 (35.29%)	9 (16.98%)	
Hypertension	no	45 (90.00%)	43 (81.13%)	0.202 *****
	yes	5 (10.00%)	10 (18.87%)	
Diabetes mellitus	no	42 (82.35%)	47 (88.68%)	0.359 *****
	yes	9 (17.65%)	6 (11.32%)	
Asthma	no	50 (96.15%)	49 (92.45%)	1 **
	yes	2 (3.85%)	4 (7.55%)	

* Pearson’s chi-square test, ** Fisher’s exact test.

**Table 5 ijerph-18-12011-t005:** Clinical course of illness.

Severity	Cases	Controls	Total
1 mild illness	9 (17.31%)	6 (11.32%)	15
2 moderate illness	25 (48.08%)	32 (60.38%)	57
3 severe illness	17 (32.69%)	14 (26.42%)	31
4 critical illness	1 (1.92%)	1 (1.89%)	2
Total	52	53	105

**Table 6 ijerph-18-12011-t006:** Clinical course parameters.

Variable		Cases	Controls	*p*-Value
Length of hospitalisation (days)	N	52	53	0.120
	Median (IQR)	8 (6)	10 (5)	
	Range	2–23	3–25	
Percentage of lung involvement on CT (%)	N	34	46	0.539
	Median (IQR)	20 (11)	27.5 (35)	
	Range	1–60	0–80	
Time from the onset of SARS-CoV-2 symptoms (days)	N	51	50	0.863
	Median (IQR)	7 (6)	7.5 (6)	
	Range	1--23	2–21	
**Oxygen flow (L/min)**	N	52	53	**0.009**
	Median (IQR)	4 (4.5)	2 (5)	
	Range	0–15	0–15	
**Variable**		**Cases**	**Controls**	** *p* ** **-Value (Pearson’s Chi-Square Test)**
Need for oxygen supplementation	no	10 (19.23%)	23 (43.40%)	0.008
	yes	42 (80.77%)	30 (56.60%)	
Need for invasive ventilation	no	52 (100.00%)	52 (98.11%)	0.32
	yes	0 (0.00%)	1 (1.89%)	
**Need for high-flow oxygen therapy**	no	43 (82.69%)	52 (98.11%)	**0.007**
	yes	9 (17.31%)	1 (1.89%)	
Need for ICU admission	no	50 (96.15%)	52 (98.11%)	0.547

**Table 7 ijerph-18-12011-t007:** Patients’ laboratory results.

Variable		Cases	Controls	*p*-Value
Hemoglobin (g/dL)	N	52	53	<0.001
	Median (IQR)	12 (1.1)	13.3 (1.6)	
	Range	7.3–15	9–16.4	
**Leukocytes (×10^3^/μL)**	N	52	53	**0.012**
	Median (IQR)	7.7 (3.23)	6.37 (3.89)	
	Range	2.99–18.58	2.12–18.13	
**Lymphocytes (×10^3^/μL)**	N	52	53	**0.002**
	Median (IQR)	0.97 (0.32)	1.35 (0.92)	
	Range	0.6–2.65	0.32–3.23	
**Neutrophils (×10^3^/μL)**	N	52	53	**<0.001**
	Median (IQR)	5.8 (3.11)	4.17 (2.98)	
	Range	2.16–16.77	1.21–14.89	
**Platelets (×10^3^/μL)**	N	52	53	**0.003**
	Median (IQR)	184.5 (58.5)	236 (104)	
	Range	34–444	0.32–475	
AST—Aspartate aminotransferase (U/L)	N	52	52	0.879
	Median (IQR)	28 (18.5)	26.5 (26.5)	
	Range	9–263	0–211	
APTT—activated partial thromboplastin time (s)	N	49	40	0.318
	Median (IQR)	33.7 (5.5)	32.45 (5.2)	
	Range	22.2–47	25.6–63.6	
ALT—Alanine aminotransferase (U/L)	N	50	52	0.10
	Median (IQR)	24 (24)	25 (33.5)	
	Range	6–177	6–130	
Bilirubin (mg/dL)	N	49	40	0.16
	Median (IQR)	0.46 (0.51)	0.36 (0.28)	
	Range	0.09–1.66	0.14–1.81	
**NT pro-BNP (pg/mL)**	N	44	34	**0.006**
	Median (IQR)	29.5 (39)	69.5 (135)	
	Range	5–260	9–1109	
**Calcium (mmol/L)**	N	43	18	**0.021**
	Mean ± SD	2.16 ± 0.09	2.22 ± 0.1	
	Range	1.86–2.41	2.05–2.48	
CK—creatine kinase (U/L)	N	50	31	0.653
	Median (IQR)	61.5 (69)	60 (62)	
	Range	11–729	15–1019	
**CRP—C Reactive Protein (mg/L)**	N	52	53	**<0.001**
	Median (IQR)	46.95 (52.4)	18 (45.5)	
	Range	1.2–160	0.3–210.4	
**D-dimer (μg/L FEU)**	N	32	47	**<0.001**
	Median (IQR)	1178 (515.5)	624 (601)	
	Range	458–4235	211–6650	
**Ferritin (ng/mL)**	N	50	21	**0.001**
	Median (IQR)	83.5 (109)	222 (204)	
	Range	13–450	0–699	
**Fibrinogen (mg/dL)**	N	50	20	**<0.001**
	Median (IQR)	553 (225)	363.5 (230.5)	
	Range	273–941	186–654	
Glucose (mg/dL)	N	41	24	0.421
	Median (IQR)	92 (17)	94 (33)	
	Range	67–157	72–253	
Potassium (mmol/L)	N	52	50	0.163
	Median (IQR)	3.93 (0.64)	4.02 (0.63)	
	Range	2.95–4.96	3.33–4.57	
**Creatinine (mg/dL)**	N	50	50	**<0.001**
	Median (IQR)	0.52 (0.17)	0.74 (0.18)	
	Range	0.19–0.9	0.51–1.7	
LDH (U/L)	N	50	42	0.782
	Median (IQR)	226.5 (91)	237 (118)	
	Range	103–389	142–507	
**Magnesium (mg/dL)**	N	46	21	**<0.001**
	Median (IQR)	1.74 (0.16)	2.15 (0.45)	
	Range	1.41–3.43	1.7–2.71	
**Sodium (mmol/L)**	N	52	49	**<0.001**
	Median (IQR)	138 (4)	141 (3)	
	Range	134–144	120–147	
**Procalcitonin (ng/mL)**	N	51	47	**<0.001**
	Median (IQR)	0.14 (0.19)	0.06 (0.09)	
	Range	0.03–95.5	0.02–15	
**PT—Prothrombin time (s)**	N	51	42	**<0.001**
	Median (IQR)	1 (0.08)	1.17 (0.16)	
	Range	0.891.29	0.97–1.75	
**Urea (mg/dL)**	N	52	47	**<0.001**
	Median (IQR)	11 (5)	22 (11)	
	Range	4–20	12–75	
Vitamin D3 25(OH) (ng/mL)	N	45	24	0.776
	Mean ± SD	27.15 ± 10.82	28.08 ± 16.31	
	Range	6.8–56.3	0–62.5	
Anti-SARS-CoV-2 IgG (ELISA) (index)	N	49	2	0.596
	Median (IQR)	3.8 (4.1)	7.35 (7.7)	
	Range	1.8–78.1	3.5–11.2	
Anti-SARS-CoV-2 IgA + IgM (ELISA) (index)	N	49	2	0.056
	Median (IQR)	0.8 (2.8)	21.85 (36.3)	
	Range	0.1–40	3.7–40	
**IL-6 (pg/mL)**	N	52	29	**<0.001**
	Median (IQR)	26.5 (31.6)	5.06 (7.79)	
	Range	1.5–113	0–70	
Troponin I (ng/mL)	N	52	31	0.071
	Median (IQR)	2.75 (2.15)	1.80 (2.7)	
	Range	0–25	0–3.4	

## Data Availability

The datasets generated and/or analyzed during the current study are not publicly available due to our policy but are available from the corresponding author upon reasonable request.

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
