# Peer review of "Is the Course of COVID-19 Different during Pregnancy? A Retrospective Comparative Study"

_ijerph, 2021, doi:10.3390/ijerph182212011_

Round 1

Reviewer 1 Report

As this is a current issue, I congratulate the authors on their initiative.

I only leave one suggestion for improvement: I suggest a brief review of the literature on the topics addressed following the introduction. (line 103).

Author Response

Brief review of the literature is added in lines 105-113

Reviewer 2 Report

The submitted manuscript studied Covid infection in pregnant patients and found that Covid caused more severe pathophysiological changes in pregnant patients compared to Non-pregnant patients.

Major

Fisher's exact test should be applied when the expected number of frequencies in the chi-square test is less than 5.

Minor

L147 "the Chi2 test" should be written as "chi-squared test" or "χ² test."

Author Response

Fisher's exact test should was applied when the expected number of frequencies in the chi-square test was less than 5 - correction in table 3 and 4

"the Chi2 test" was corrected with  "chi-squared test" - in line 175

Reviewer 3 Report

This manuscript by Januszewski et al. compared the clinical course and laboratory tests between pregnant women and respective controls. This manuscript contains useful information for readers. However, the authors need to address some concerns, which need to be clarified before publication in IJERPH. Please see my comments, explained below.

Major comments:

  1. Please discuss and/or provide some insights about the influence of vaccination against pregnant women, especially the difference of dosing timing.

Author Response

The discussion about influence of vaccination against pregnant women  and the difference of dosing timing was added in lines 250-260

This manuscript is a resubmission of an earlier submission. The following is a list of the peer review reports and author responses from that submission.